# Enhancing Inhibitory Control in Older Adults: A Biofeedback Study

**DOI:** 10.3390/brainsci13020335

**Published:** 2023-02-15

**Authors:** Doriana Tinello, Mika Tarvainen, Sascha Zuber, Matthias Kliegel

**Affiliations:** 1Faculty of Psychology and Educational Sciences, University of Geneva, Boulevard du Pont d’Arve 28, 1205 Geneva, Switzerland; 2LIVES, Overcoming Vulnerability, Life Course Perspective, Swiss National Centre of Competence in Research, University of Lausanne, Géopolis Building, 1015 Lausanne, Switzerland; 3Centre for the Interdisciplinary Study of Gerontology and Vulnerability (CIGEV), University of Geneva, Boulevard du Pont d’Arve 28, 1205 Geneva, Switzerland; 4Department of Technical Physics, University of Eastern Finland, 70211 Kuopio, Finland; 5Department of Clinical Physiology and Nuclear Medicine, University of Eastern Finland and Kuopio University Hospital, 70029 Kuopio, Finland; 6Institute on Aging & Lifelong Health, University of Victoria, Victoria, BC V8W 2Y2, Canada

**Keywords:** heart rate variability, biofeedback, hemoencephalography neurofeedback, inhibitory control, older adults

## Abstract

Multidomain interventions based on bio-/neurofeedback have proven useful in improving executive functions. The present study aimed to explore the potential efficacy and feasibility of an intervention that combined Heart Rate Variability Biofeedback (HRV-BF) and Near Infrared Hemoencephalography Neurofeedback (nirHEG-NF) on inhibitory control (IC) of healthy older adults. Thirty-four participants were randomly assigned to two groups: the biofeedback group (received a 10-week combined intervention of HRV-BF and nirHEG-NF) and the active control group (received a similar protocol without real-time biofeedback). Besides cognitive outcomes, the study examined pre- and post-changes in autonomic regulation and prefrontal blood oxygenation at rest and during training. Results revealed training-induced inhibitory control gains in one of the two interference tasks, whereas no effect was found on response inhibition. After the intervention, HRV increased in participants with the lowest levels of HRV at baseline. Although older adults increased blood oxygenation during training, no significant pre- and post-changes were found in blood flow oxygenation. These findings not only suggest that HRV-BF and nirHEG-NF potentially improve performance in certain subcomponents of inhibition (i.e., interference vs. response inhibition), but it may also be beneficial for parasympathetic activity in participants with low HRV and for increasing blood flow oxygenation on prefrontal areas during training.

## 1. Introduction

Inhibitory control (IC) is a pivotal mechanism underlying executive functions and consists of two main processes: response inhibition and interference control [1,2,3]. Response inhibition refers to the ability to suppress prepotent responses, inappropriate actions and emotions, resist temptations, or give impulsive responses [2,4]. Interference control relies on selective attention and involves the ability to downregulate sattentional processes oriented toward irrelevant information or resisting distractions in the environment [1,5,6,7,8]. Interference control is also related to the capacity to disengage attention from negative thoughts or stimuli or resist retrieving negative memories from the past (i.e., negative priming) [2,9].

While the hypothesis of a general inhibition deficit in older adults is currently under debate, it is clear that at least some components of IC are compromised in old age (see Rey-Mermet & Gade for a meta-analysis [3]). More importantly, for present purposes, the decline of IC in old adulthood has been described as critical for older adults’ everyday life because it may severely compromise independence and quality of life [10]. Neuroimaging studies provide strong evidence that supports the association between different components of inhibitory control and related brain areas. These studies found that different types of interference tasks were related to different neural networks. For example, Stroop was associated with the right prefrontal cortex (PFC), while the Flanker task recruited the left PFC, and the frontoparietal network was implicated in action withholding [11,12].

Less clear are the age-related differences in this association. According to Munakata et al. [13], inhibitory control is rooted in two neural mechanisms. Indirect-competitive inhibition selects goal-relevant responses and suppresses irrelevant ones (i.e., Stroop and Flankers tasks). In contrast, direct-global inhibition involves the suppression of an initiated action (i.e., Stop-signal task). Previous studies using neuroimaging investigations showed evidence of an age-related decrease in performance in attentional control tasks requiring indirect-competitive inhibition and a compensatory activity of the left prefrontal cortex in addition to other areas active in young adults [14]. Coxon et al.’s study [15] found that while young adults activated the right inferior frontal cortex (rIFC), the pre-supplementary motor area (preSMA), and basal ganglia nuclei when performing a stop-go signal task; older adults presented a different pattern. Overall, older adults showed less effective recruitment of the same regions but a more distributed activity of the occipital and parietal lobes. In addition, white matter connectivity in the areas involved in inhibitory control was associated with increased activation of preSMA, suggesting that performing cognitive tasks may contribute to preserving brain functions (Coxon et al., 2016). Researchers have shown a growing interest in developing interventions to prevent cognitive decline in older adults [16]. So far, cognitive training (CT) aimed to preserve, maintain or enhance cognitive functioning in the older population has been the most studied approach [16,17,18]. Yet, the benefits of such methods are controversial. Recently, in a Cochrane review, Gates et al. [16] concluded that computerized CT shows insufficient evidence to support beneficial effects on global cognition, memory, speed of processing, or executive functions of older adults. The review suggests that the lack of transfer and generalization effects may be the two main drawbacks. In recent years, an increasing number of studies indicate that broader, multidomain interventions aimed to compensate for cognitive decline or mild cognitive impairment would be more beneficial [19,20,21,22]. These interventions typically involve techniques targeting different domains, such as physical activity, dietary changes, cognitive training, and social engagement. For example, some meta-analyses have shown that physical exercise in conjunction with CT produces positive results in cognitive functions of older populations [23,24,25]. In particular, a recent systematic review and meta-analysis of 50 studies showed that performing physical activity and CT simultaneously had a larger effect on executive functions, speed, and global cognition in healthy older adults [26]. In addition, findings showed that aerobic training had a more beneficial effect on attention and fitness, while non-aerobic training produced larger effects on general cognition and balance. Based on neuroscientific methods, neurofeedback (NF) and heart rate variability biofeedback (HRV-BF) have shown the first promising results in improving cognitive functioning in older populations [27,28,29,30,31]. The use of different types of biofeedback techniques to enhance executive functioning is rooted in the assumption that these methods can influence brain functional hemispheric asymmetry [32]. For example, one study showed that electromyographic biofeedback to reduce stutter verbal behavior in adult males led to covariation between verbal fluency and hemispheric alpha asymmetry [33]. Au et al. [34] found that neurofeedback training of the sensorimotor cortex could increase the efficiency of inhibitory control in children with developmental dyslexia. Another study that employed neurofeedback alpha training showed evidence for a concomitant increase in the upper alpha band and short-term memory performance [35]. While a study with young athletes found that heart rate variability biofeedback could induce an increase in frontal theta power (associated with cognitive processing), a reduction of frontal EEG asymmetry in alpha power (associated with a decreased arousal), as well as a shift toward parietal and occipital asymmetry during eyes-open associated to increased feelings of well-being) [36].

In the context of multidomain interventions targeting older adults, Meeuwsen et al. [37] explored the benefits of a program combining HRV-BF, NF, physical exercise, diet, mindfulness, and face-to-face coaching on participants with Mild Cognitive Impairment and subjective memory complaints. The authors used a 3-month waitlist control period before the beginning of the study as a control condition. They found significant improvements over the treatment period on the MoCA test assessing visuospatial/executive, naming, attention, language, abstraction, delayed recall, and orientation [38], as well as in several self-reported measures of mental health and quality of life. Concerning physiology, while this study showed significant long-lasting improvements in brain oscillations after the treatment (as assessed after a 6-month follow-up), it found no evidence to support the effect of the intervention on HRV parameters. Thus far, literature has paid little attention to the effectiveness of HRV-BF in improving cognitive functions of older adults. In a recent review, we showed that only nine studies (out of the 16 included) targeting old populations showed beneficial effects of HRV-BF on attentional skills but had no impact on cognitive flexibility and interference control [31,39]. However, several limitations weaken the evidence of gains due to the specificity of the intervention: the small sample size, the lack of a control group, and the absence of HRV parameters assessment. Another relevant field of research closely related to NF is Brain–Computer–Interface–Technology (BCI). So far, BCI techniques have been regarded as non-invasive, innovative methods employed for rehabilitation purposes and improving cognitive functioning [40]. There is scientific evidence to support the use of BCI technology in implementing NF for cognitive enhancement of healthy adults. For example, Thomas and Vinod [41] showed that NF training applied to BCI games could enhance attention and cognitive skills performance of game players. Another study including old adults showed that a combined intervention of NF and motor-imagery-based BCI could improve four cognitive functions in older adults: visuospatial, language, memory, and intellectual. The authors concluded that BCI-based NF might be regarded as a promising strategy to counteract age-related cognitive decline [42].

The present study is the first to assess the impact of a multimodal biofeedback intervention combining HRV-BF and nirHEG-NF on IC of healthy older adults. The current trial sought to fill the gaps in the existing literature by (a) specifically measuring different sub-components of the same domain (i.e., IC) via three different tasks; (b) including an active control group (CTRL) exposed to a similar set-up but without feedback; (c) measuring the impact of the intervention on parasympathetic activity and prefrontal blood flow oxygenation. The rationale for those techniques is presented below.

### 1.1. Heart Rate Variability Biofeedback for Cognitive Enhancement

HRV-BF is a non-invasive technique that provides individuals with real-time information about their HRV to help them learn to regulate their autonomic nervous system (ANS) [43]. HRV-BF has received growing interest because of its effectiveness in altering heart rate variability and positively influencing physical wellbeing [44,45]. HRV refers to the changes in the time interval between two consecutive heartbeats [46]. The neurovisceral integration model proposes a theoretical framework within which HRV, cognition, and emotions share common neural structures and contribute to an organism’s self-regulation and adaptability [47,48]. In this model, a network of cortical and subcortical areas is related to autonomic functions (e.g., cardiovascular, respiratory, and visceromotor activity) via sympathetic and parasympathetic pathways. Some of the key components of this network include the insula, the medial prefrontal cortex, the amygdala’s central nucleus, and the tractus solitarius, which receives information from the autonomic nervous system carried by the vagus nerve and projects to other brain regions [49]. Thayer et al.’s [47] main findings showed that (a) higher resting HRV levels are associated with higher performance in executive functions and that (b) HRV can be altered by behavioral programs so that the modulation of HRV may affect cognitive functions [50,51].

On the other hand, reduced levels of HRV have been associated with a reduced cardiac regulatory capacity [52]; obsessive-compulsive drinking behavior [53]; high levels of cholesterol [54], hyperglycemia [55], and hypertension [56]; and an increased risk of mortality [45]. During HRV-BF, participants learn to modify their HRV through slow-paced breathing (e.g., six cycles per minute). This breathing rhythm stimulates the vagal nerve and increases heart rate oscillations improving physical, emotional, and cognitive functions [44]. In a recent review, we cdocumented how HRV-BF may improve executive functions across the lifespan. Our findings suggest that this method might be a promising candidate to improve cognition, particularly among more vulnerable populations (e.g., individuals exposed to stressful environments or with lower performance on baseline cognitive measures) [31]. 

### 1.2. Near Infrared Hemoencephalography Neurofeedback 

NirHEG-NF is a non-invasive technique that provides individuals with real-time information about their brain blood oxygenation [57]. NirHEG-NF is a particular form of neurofeedback that uses near-infrared spectroscopy and relies on the principle of neurovascular coupling. This means that changes in neural activity in a given cortical region are reflected by changes in capillary blood oxygenation in that region, with an increase in oxygenated-hemoglobin concentration (O2Hb) when the energy demand of the tissue increases [44,45]. An example of the application of this approach is given from a recent study that proposed a model of assessment of Attention Deficit Hyperactivity Disorder (ADHD) based on the relationship between activation of selected areas of the brain—measured with nirHEG and Electroencephalography (EEG), and differences in performance on various aspects of executive functioning [58]. To date, the few existing studies that used nirHEG-NF proved to be effective in treating different conditions, such as ADHD disorder [57,59,60], obesity [61], and schizophrenia [62]. However, to our best knowledge, no studies have examined the impact of this technique in combination with HRV-BF on the executive functions of healthy older adults. 

### 1.3. Objectives 

Based on the above reasoning, the primary objective of this study was to assess the potential efficacy and feasibility of a combined intervention of 10-week HRV-BF and nirHEG-NF on IC of healthy older adults. Moreover, the study explored the impact of the intervention on pre-post differences in HRV values and prefrontal blood oxygenation at rest and during training. The experimental group was compared to a CTRL group exposed to the same environment and connected with the same sensors but without the feedback of their physiological activity. The research hypotheses were as follows: (a) the intervention group (BF) will show greater pre-post changes in the sub-components of inhibitory control (interference control and response inhibition) than the control group (CTRL); (b) the BF group will show higher gains in HRV values than the CTRL group, and (c) the BF group will show higher prefrontal blood oxygenation changes at rest and during training than participants in the CTRL group.

## 2. Materials and Methods

### 2.1. Study Design

We designed a 10-week, randomized, controlled study to examine the potential efficacy and assess the feasibility of a combined biofeedback intervention on inhibitory control of older adults. The study was planned for 50 participants. However, due to the COVID pandemic restrictions, we could not obtain the targeted number of participants, and the final sample included 34 participants.

### 2.2. Participants

In this study, participants were recruited by advertisements from September 2018 to February 2021. Inclusion criteria included: age 65–80 at the start of the trial, good general health, and being native speakers or fluent in French. Exclusion criteria included: past or current history of neurological disorder, cerebrovascular disease or heart failure, current treatment (or within the last 4 weeks) with antipsychotics or benzodiazepines medication. The final sample consisted of 34 participants (23.5% men); the mean age was 70.84, *SD* = 4.07; the mean education was 15.41 years, *SD* = 3.88. Participants preliminarily screened were randomized to either the experimental group (BF: *N* = 19; *M_age_* = 70.78, *SD* = 3.97; *M_edu_* = 14.89, *SD* = 3.59) or the control group (CTRL: *N* = 15; *M_age_* = 70.92, SD = 4.33; *M_edu_* = 15.93, *SD* = 4.27). The research protocol was approved by the Cantonal Ethical Committee of the Canton of Geneva and was conducted following the 1964 Helsinki declaration and its later amendments or comparable ethical standards. All participants were compensated with 100 CHF for their participation. The study is registered on ClinicalTrials.gov ID: NCT04925830.

### 2.3. Procedure

Screening surveys (F-TICS and demographic questionnaire) were used to gather information on inclusion-exclusion criteria [63]. For the F-TICS, a cut-off of 37/43 is recommended to exclude individuals with mild cognitive impairment (MCI) [64]. Within 2 weeks of the screening surveys, participants who enrolled in the study returned to the laboratory centre to complete the baseline assessment. The following week, all participants underwent a physiology evaluation in which HRV and prefrontal blood flow oxygenation were measured. Next, the intervention for the BF group consisted of ten 25-min HRV–BF sessions followed by 30-min nirHEG-NF sessions, once a week. The CTRL received the same sensors and was exposed to the same videos as the BF group, but the biofeedback option was disabled. Within 1 week from the end of the intervention, both groups completed the same cognitive and physiological evaluation as at baseline. The experimental procedure is described in Figure 1.

### 2.4. Pretest Cognitive Assessment

In this study, computerized tasks (Arrows and Go/no Go) were implemented in E-Prime 2.0. Two tasks were used to appraise interference control: the Arrows task [53] and the Stroop task [54]. The Go/no Go task [55] was used to assess response inhibition. 

#### 2.4.1. Arrows Task

In this task, a white point appeared in the middle of the screen for 500 ms, succeeded by an arrow pointing to the right or to the left (500 ms). Participants had to decide as quickly as possible the direction of the arrows by pressing the key to the right or left of the keyboard. Thus, the task consisted of three types of trials: congruence trials, in which the place of appearance of the arrow and its direction corresponded (the arrow pointing to the right appeared on the right on the screen); the non-congruent trials, in which the place of appearance of the arrow and its direction did not match (the arrow pointing to the right appeared on the left); the neutral trials when the arrow appeared in the centre of the screen. The difference between the congruent and non-congruent trials is called the interference effect and reflect the cost of activating an irrelevant response inhibition mechanism to give the relevant answer [65]. Our task consisted of two blocks. The first block was made of 18 practice trials (6 neutral, 6 congruent, and 6 incongruent trials). The second block consisted of 48 trials (16 neutral, 16 congruent, and 16 incongruent). Then, the interference score was given by interference reaction time (RT). The task lasted for approximately 5 min.

#### 2.4.2. Go/No Go Task

In the Go/no Go task, for each trial, a number was presented in the centre of a black screen for up to 900 ms. Each number was preceded by a fixing cross, which was presented for 1000 ms. Participants were instructed to answer as fast as possible to every number by pressing the “down arrow” key (Go signal) but not to answer when the number was 3 (no Go signal). The frequency of this no Go signal was set at 10% (number 3 appeared 12 times out of 120 trials). The task lasted for approximately 10 min. The outcomes were the average of the reaction time as a measure of processing speed and the proportion of commissions (no Go) as a quantitative measure of inhibitory control.

#### 2.4.3. Stroop Task

The Stroop task was the original colour-word paper version [66] in which participants were asked to read three different tables of 36 stimuli as fast as possible. The first one represents the “neutral condition” and requires participants to read words of colours printed in black ink. The second table represents the “congruent condition” and requires participants to name different colour squares (e.g., “blue” shown in blue). The third table represents the “incongruent condition” and requires participants to name the colour of the ink of colour-words printed in an inconsistent colour ink (e.g., “red” written in blue) [67]. The measure of the interference of conflicting word stimuli upon naming colours, the Stroop effect [66], is the difference in the time for naming the colours in which the words are printed in the incongruent condition and the colours printed in squares of the congruent condition. The interference score (in seconds) was computed by subtracting the speed of performance in the incongruent condition from the speed of performance in the congruent condition.

### 2.5. HRV and NirHEG Pretest Assessment

All participants were instructed to refrain from performing physical exercise or consuming any caffeine, alcohol, nicotine, or a heavy meal for 2 h before the start of the physiological baseline measure. They were asked to sit comfortably on a chair with their eyes closed while attached to the sensors. Respiratory rate was recorded with a respiratory belt positioned around the abdomen. Heart rate was measured using a blood volume pulse sensor (BVP) positioned on the fingertip of the non-dominant hand. The BVP sensor sends an infrared light to the finger and continuously measures the intensity of light reflected by the tissue. Heartbeats modulate the intensity of the reflected light, and pulse-to-pulse intervals can be extracted from the BVP signal, providing “an accurate approximation” of the interbeat intervals or IBI [68]. After a short phase of familiarization with the sensors, the 5-min HRV baseline measures began. Next, the assessment of the participant’s baseline blood oxygenation lasted 2 min and required to silently count back from 500 at each of the three prefrontal sites measurements (Fpz, Fp1, and Fp2, according to the international 10–20 system placement [69]). All physiological measures were recorded using the NeXus-10 MKII hardware and BioTrace + software, version 2018 (Mind Media, Herten, The Netherlands). 

### 2.6. HRV-BF + NirHEG-NF Training

#### 2.6.1. BF Group

Within 2 weeks from the physiological baseline session, participants of the biofeedback group received their first combined training. The HRV-BF intervention consisted of ten25-min HRV-BF training sessions once per week. Each session started with a 5-min resting HRV recording. Participants were asked to sit as quietly as possible in a comfortable chair with knees at a 90° angle, both feet on the floor and hands on their thighs, eyes closed, and to breath at their natural rhythm (for details about the recommendations for HRV measures, see Laborde et al. [70]). Further, participants were taught to breathe at their resonant frequency, which is the frequency that maximizes HRV amplitudes [71]. Each subject’s resonance frequency was defined by measuring HR oscillation amplitudes while the participant breathed at the following paces: 6.5, 6.0, and 5.0 breaths/min for ∼2–3 min. A pacer stimulus that moved at the participant’s resonant frequency was displayed on the experimenter screen. Throughout training sessions, the participant was instructed to breathe slowly, in phase with heart change at her own resonance frequency, using abdominal or pursed lips breathing techniques with longer exhalation than inhalation. A dual screen was used, one for the physiological monitoring and one for the participant training. Participants sat in front of a 43–48 cm computer screen that showed their physiological signals and gave HRV-BF principally by means of a vertical bar graph dynamically varying in height. The bar graph was associated with a relaxing video, signaling successful regulation of one’s autonomous nervous system by forwarding motion and allowing a calming music or, on the contrary, providing negative feedback by interruption of the video and the audio. The participants of the HRV-BF group were also instructed to practice paced breathing at 6 breaths/min on a daily basis for at least 5 min (up to 20 min per day) for three days per week. They were also asked to fill a diary record detailing the date, the duration of the breathing exercises, and their experience. 

After a 10-min break from the end of HRV-BF training, participants started nirHEG-NF training for 30 min. The nirHEG apparatus consists of an electronics box and a headband (Biocomp Research Institute (Los Angeles, CA, USA)) equipped with two lights of red and infrared low frequencies (660 nm and 850 nm respectively) and a light receiver sensor, which is sensitive to the returning light. The headband, which incorporates the two light sources and the light receiver, was placed in contact with three sites of the prefrontal area: Fpz, Fp1, Fp2. The lights alternatively pass through the skin, at ~1.50 cm deep, penetrate the vascular cortical tissues, scatter, bounce, and are reflected back to the sensor. Depending on the level of blood oxygenation, the red light will be more or less absorbed by the hemoglobin. The non-absorbed light is amplified, rectified, and converted into a value representing the ratio between the two lights. Each nirHEG-NF session began with a 2-min baseline measure on each of the three prefrontal sites (Fpz, Fp1, Fp2). During this time, participants sat in a comfortable chair with knees at a 90° angle, both feet on the floor, eyes closed, and were instructed to count back one from 500 mentally. Next, nirHEG-NF were principally given using a vertical bar dynamically varying in height and represented on a computer monitor. The height of the bar changed according to the hemodynamic response measured at each of the three prefrontal points (Fpz, Fp1, Fp2). The bar graph was associated with a game-like animation, signaling successful regulation of blood oxygenation by forwarding the motion of the pieces of a puzzle or providing negative feedback by freezing the pieces of the puzzle.

#### 2.6.2. CTRL Group

Participants allocated to the CTRL group were connected to the same sensors and received the same number of sessions for the same frequency and duration as the BF group, but the bar graph feedback on their screen was concealed. We provided articles addressing health and aging topics as a home exercise.

### 2.7. HRV Measures

Conventionally, HRV has been quantified by several parameters and different methods; a detailed description is available elsewhere [70]. In the context of this study, we selected two time-domain HRV measures: the standard deviation of all normal-to-normal (NN) intervals (SDNN) and the root mean square of the successive R-R intervals (RMSSD). While SDNN describes the overall HRV, RMSSD is mainly related to respiratory sinus arrhythmia (i.e., heart rate increases during inspiration and decreases during expiration) and parasympathetic activation [72]. However, although we collected frequency domain components, of HRV (i.e., LF power and HF power), we did not perform power spectral analyses because the frequency domain analyses become unreliable when respiratory rates overlap with the low-frequency band (0.04–0.15 Hz). A reliable assessment of LF and HF component powers would require the respiratory rate within the HF band (0.15–0.4 Hz) or a fixed breathing rate to be applied [73]. In order to avoid to further reducing the analyses’ power due to the relatively small sample size, we chose not to exclude cases whose respiratory rate overlapped with the LF bands and instead opted to perform time domain analyses (i.e., RMSSD, SDNN). Further, to assess change in HRV variables, we subdivided low versus high HRV groups based on the median split on the log-transformed RMSSD at baseline. HRV parameters were calculated from the inter beat intervals (IBI) data utilizing the Kubios HRV Analysis software (Kubios Oy, Kuopio, Finland). 

### 2.8. NirHEG Measures

To measure blood flow oxygenation values at rest, we extracted the HEG ratio, representing the ratio between red and infrared light × 100. Then, to evaluate the change in blood oxygenation during the training, we calculated the HEG gain on three-time points: at the beginning of the program (first session), in the second half of the program (the seventh session), and at the end (tenth session). HEG gain was obtained by applying the following formula: 

Mean HEG value during measurement/mean HEG value from the first 10 s in the segment −1 [57,74]. 

### 2.9. Statistical Analyses

Statistical analyses were performed using version 25 of the Statistical Package for Social Science (SPSS 25, Chicago, IL, USA). To investigate the effect of the combined intervention of HRV-BF and HEG-NF on inhibitory control, we performed several 2 [time (pre, post)] × 2 [groups (BF, CTRL)] repeated measure ANOVAs, with time as within-subject factor and group as between-subject factor. In addition, we computed *t*-tests to assess within and between-group differences at individual time points. To investigate the effect of the intervention on HRV metrics, we computed 2 [time (pre, post)] × 2 [groups (BF, CTRL)] × 2 [RMSSD level (low RMSSD, high RMSSD)] repeated measure ANOVAs, with time as within-subject factor, group, and RMSSD level as between-subject factors. To evaluate the effect of the intervention on superficial prefrontal blood oxygenation at rest, we performed 2 [time (pre, post)] × 3 [sites (Fpz, Fp1, Fp2) × 2 [groups (BF, CTRL)] repeated measure ANOVAs, with time and site as within-subject factors, and group as between-subject factor. Finally, to assess the differences in HEG gains during sessions, we computed 3 [time (first, seventh, tenth)] × 3 [sites (Fpz, Fp1, Fp2) × 2 [groups (BF, CTRL)] repeated measure ANOVAs, with time and site as within-subject factors, and groups as between-subject factor. Greenhouse–Geisser Epsilon corrections were used when sphericity was violated. Significant *p*-value was set at *p* < 0.05. The effect size was assessed based on partial η^2^ or Cohen’s d, and Bonferroni correction was applied for multiple comparisons.

## 3. Results

### 3.1. Baseline Characteristics

Table 1 shows the descriptive statistics of demographic, cognitive, and physiological variables. We used Shapiro–Wilk’s test to assess the normality of distributions. Due to the skewed distributions, HRV variables were log-transformed. No Go error commissions were affected by the floor effect.

### 3.2. Training Effects on Interference Control

#### 3.2.1. Arrows Task

A repeated measures ANOVA revealed a statistically significant interaction between group and time (*F*(1,32) = 4.84, *p* = 0.035, $\eta_{p}^{2}$ = 0.131). There was no significant effect of time, *F*(1,32) = 1.33, *p* = 0.258, $\eta_{p}^{2}$ = 0.040, neither of group, *F*(1,32) = 0.21, *p* = 0.653, $\eta_{p}^{2}$ = 0.006. Further, paired sample *t*-test showed lower reaction time for participants in the BF group at the post test (98.358 ± 53.398) compared to the pre-test (145.960 ± 64.619), (*t*(18) = −2.43, *p* = 0.026, *d* = −0.557). There was no significant change in reaction time between the pre-test (122.457 ± 54.397) and the post-test (137.316 ± 82.604) for participants of the CTRL group (*t*(14) = 0.74, *p* = 0.472, *d* = 0.191) (Figure 2).

#### 3.2.2. Stroop Task

A repeated measures ANOVA revealed that there was no significant interaction between group and time (*F*(1,32) = 2.64, *p* = 0.114, $\eta_{p}^{2}$ = 0.076). There was a significant effect of time (*F*(1,32) = 10.52, *p* = 0.003, $\eta_{p}^{2}$ = 0.247). Post-hoc Bonferroni-corrected comparisons indicated that response time was higher at posttest compared to the pretest. However, there was no significant effect of group (*F*(1,32) = 1.04, *p* = 0.315, $\eta_{p}^{2}$ = 0.031). Further, paired sample *t*-tests revealed that participants of the CTRL group increased time reaction at post-test (23.814 ± 11.320) compared to the pre-test (17.704 ± 8.370), (*t*(14) = 3.37, *p* = 0.005, *d* = 0.871). There was no significant change in response time from the pre-test (17.073 ± 7.704) to the post-test (19.105 ± 6.147) for participants of the BF group (*t*(18) = 1.19, *p* = 0.251, *d* = 0.272) (Figure 3).

### 3.3. Training Effects on Response Inhibition

#### Go/No Go Task

As a result of the analysis, there was no significant effect of time, *F*(1,32) = 0.78, *p* = 0.383, $\eta_{p}^{2}$ = 0.024, neither of group, *F*(1,32) = 0.03, *p* = 0.854, $\eta_{p}^{2}$ = 0.001, nor interaction effects (*F*(1,32) = 0.23, *p* = 0.632, $\eta_{p}^{2}$ = 0.007). With regard to the no Go proportion of errors, overall, both groups were highly accurate, with correct no Go trials nearly approaching the ceiling. The positively skewed distribution required the use of non-parametric statistics. Thus, a Mann–Whitney U test was run to determine if there was a difference in the no Go proportion of error distribution at the post-test between the two groups. Results revealed that distributions and medians were not statistically significantly different (*U* = 140, *p* = 0.940). We conducted the Wilcoxon signed rank test to test within-group differences from pre- to post-test. The distributions of the proportion of errors for the BF and CTRL groups were similar, as assessed by visual inspection. There was no significant difference in the proportion of error commissions in experimental participants between the pre- (0.162 ± 0.129) and the post-test (0.167 ± 0.121), (*Z* = −0.04, *p* = 0.963). Likewise, there was no significant difference in the proportion of error commissions among participants control between the pre-test (0.200 ± 0.150) and the post-test (0.150 ± 0.090) (*Z* = −1.61, *p* = 0.107). 

### 3.4. HRV Measures

In the first step, we performed a median split of ln RMSSD measured at the baseline to subdivide the sample into two subgroups, high RMSSD (where parasympathetic activation dominates) and low RMSSD (where sympathetic activation dominates). Then, we conducted a three-way mixed ANOVA to evaluate the effects of group, baseline HRV level (high vs. low RMSSD), time, and their interaction on RMSSD. The results of the analysis revealed a statistically significant interaction between time and baseline HRV level (*F*(1,30) = 15.38, *p* < 0.001, $\eta_{p}^{2}$ = 0.339). No other interactions were significant (*p* > 0.05). Further, paired *t*-tests revealed that when HRV level was low at baseline, ln RMSSD significantly increased after the intervention in the BF group (*t*(7) = 2.95, *p* = 0.021, *d* = 1.042), as opposed to the CTRL group (*t*(8) = 1.59, *p* = 0.150, *d* = 0.530). Instead, when HRV was high at the baseline, it tended to decrease after the intervention for both the BF and the CTRL group without reaching within-group significance: (*t*(10) = −1.70, *p* = 0.119, *d* = −0.514); (*t*(5) = −1.71, *p* = 0.147, *d* = −0.700) (Figure 4). 

Next, a three-way mixed ANOVA was run to evaluate the effects of group, baseline HRV level (high vs. low RMSSD), time, and their interaction on SDNN (a marker of global autonomic regulation). There was a statistically significant interaction between time and baseline HRV level (*F*(1,30) = 15.52, *p* < 0.001, $\eta_{p}^{2}\;$ = 0.341). No other interaction was significant (*p* > 0.05). Further, a *t*-test for the paired sample revealed that when HRV level was low at baseline, ln SDNN significantly increased after the intervention in the BF group, while similar results were not observed in the CTRL group (*t*(7) = 3.78, *p* = 0.007, *d* = 1.336); (*t*(8) = 1.77, *p* = 0.116, *d* = 0.588). Instead, when HRV was high at the baseline, ln SDNN tended to decrease after the intervention for both the BF and the CTRL group without reaching within-group significance (*t*(10) = −0.82, *p* = 0.432, *d* = −0.247); (*t*(5) = −2.21, *p* = 0.078, *d* = −0.904) (Figure 5).

### 3.5. NirHEG Ratio

We assessed pre-post HEG ratio differences between groups across the three sites using three-way mixed repeated measures ANOVAs. As a result, we found a statistically significant effect of the site (*F*(1.40,60) = 15.02, *p* < 0.001, $\eta_{p}^{2}$ = 0.459), Greenhouse-Geisser correction was applied due to the lack of variance sphericity. However, there was no significant effect of group (*F*(1,30) = 0.01 *p* = 0.912, $\eta_{p}^{2}$ = 0.00), neither of time (*F*(1,30) = 1.08, *p* = 0.306, $\eta_{p}^{2}$ = 0.035), nor were the interactions significant. Post hoc test with the Bonferroni correction applied revealed that the effect of the site factor was that participants exhibited a higher HEG ratio at Fpz compared to Fp1 and Fp2 (all *p* = < 0.001). Table 2 shows pre-and post-means and standard deviations in mean HEG ratio values by site and by group.

### 3.6. NirHEG Gain

To evaluate the impact of the intervention on HEG gain, we performed a three-way repeated measures ANOVA. The results of the 3 × 3 × 2 ANOVA with time (1st, 7th, and 10th) and site (Fpz, Fp1, and Fp2) as within-subjects factors and the group as between-subjects factor yielded a statistically significant effect of the group (*F*(1,25) = 9.46, *p* = 0.005, $\eta_{p}^{2}$ = 0.275). Post-hoc Bonferroni-corrected comparisons showed that the BF group had significantly higher gains than the CTRL group (*p* = 0.005). There was a significant effect of site (*F*(2,50) = 5.21, *p* = 0.009, $\eta_{p}^{2}$ = 0.173). Post-hoc Bonferroni-corrected comparisons revealed that HEG gains were higher at the Fpz location than Fp2 (*p* = 0.012). Neither the main effect of time nor the interactions were significant. Further, we conducted a series of one-way ANOVAs with the group as a factor on HEG gain values for each location and session. Results are displayed in Table 3 and Figure 6.

## 4. Discussion

The current study is the first to examine the effect of an intervention combining HRV-BF and nirHEG-NF on IC of healthy older adults. Previous literature shows that different biofeedback types may positively influence brain functional hemispheric asymmetry [32]. The effectiveness of biofeedback interventions is related to the possibility for individuals to learn to modulate brain activation and physiological activity to improve cognitive and emotional functioning.

Our findings showed that participants assigned to HRV-BF and nirHEG-NF training became less susceptible to interference by significantly decreasing RTs (only) in the Arrows task after the intervention. The analysis of the Stroop performance showed different results. While the CTRL group revealed significantly slower RTs at post-test, we found no pre-to-post changes in interference scores in the BF group. Boutcher and Boutcher [75] suggest that the motor response involved in the traditional verbal Stroop task may lead to increased cardiovascular activity and that the test-retest effect does not influence this task. Thus, the CTRL group may have been unable to down-regulate the autonomic activation and the negative affect associated with performing a task under pressure, leading to the increased time to respond to the conflicting stimuli. In comparison, the BF group may have better self-regulated their physiology, which may have helped them to avoid a deterioration in their performance at the post-test. Concerning the lack of a positive effect of the intervention, it is possible that performing the Stroop with an oral response modality (compared to a computerized version with a button-press response) while interacting with the experimenter may have increased the difficulty of the task. In this regard, Penner et al. [76] showed that different variants of the Stroop task (i.e., paper versus computerized version) might increase interference. When they compared the conventional color-word version to two computerized Stroop tasks, they found that the strongest effect was in the conventional color-word version, where the response modality is oral instead of manual (button presses). Thus, it is possible that in our study, the intervention effect would have been detectable if the Stroop had been administered in a computerized version. Regarding response inhibition, we did not find differences between the groups in the Go RTs trials or in no-Go errors where accuracy approached a ceiling effect. It is possible that participants may have given privilege to slower responding, allowing more accurate response and inhibition of the prepotent Go response, which is known as the compensatory mechanism of age-related speed/accuracy trade-off [77].

The tasks administered in the current study required the suppression of prepotent responses, whether triggered by verbal (Stroop effect) or spatial (Simon effect) distractors or required withholding responses (no Go trials) [11]. Neuroimaging studies have proven that these components rely on shared and distinct brain areas [12]. For example, response inhibition is lateralized on the right PFC, while interference control is lateralized on the left side [11]. However, both processes are associated with the network of brain structures that overlap with the neurovisceral integration network (e.g., anterior insula, dorsolateral prefrontal cortex, dorsal premotor cortex, preSMA) and are, thus, sensitive to HRV modifications [78]. Therefore, the variability in our results may be explained by the complex relationship between the effect of biofeedback on different components of inhibitory control and the fluctuation of hemispheric asymmetry in the aging brain. With regard to the potential benefits of the intervention on HRV parameters, the subgroup findings should be considered exploratory, given that the sample size is not large enough to detect realistic subgroup effects. In this study, subjects with lower HRV levels at baseline significantly benefitted from the intervention by increasing their parasympathetic/vagal activity and global autonomic functioning, as reflected by higher RMSSD and SDNN values. In secondary analyses (non-reported in this paper), we performed correlation analyses to assess the association between RMSSD at baseline and interference performance at baseline, as well as between RMSSD changes and interference changes in both the BF and the CTRL groups. Overall results revealed no statistically significant correlations between HRV and interference performance. The present findings contrast with previous reports supporting a positive relationship between measures of executive functioning and HRV parameters [5,79]. However, one explanation for these contradictory results could be related to the fact that our study targeted a much older population.

HRV–BF studies on elderly cognitive functions are sparse and show mixed results [31]. In the field of BF literature, studies that focused on the cognitive enhancement of older adults explored the impact of HRV-BF or NF as stand-alone techniques. Previously, Jester et al. [39] found that older adults with psychiatric symptoms could significantly improve their attentional skills but not flexibility or inhibitory control after an HRV–BF intervention. However, their study did not involve a control group and did not assess HRV parameters, thus limiting the comparability of their results. Also, in a study involving young healthy adults, Schumann et al. [80] found no group differences in inhibition outcomes between the intervention and the active control group. However, their study revealed that subjects with lower HRV levels at baseline profited more from the biofeedback training in terms of RMSSD increase than subjects with higher HRV levels at baseline. Further, Sutarto et al. [81] realized one of the first studies to assess the impact of HRV biofeedback training on cognitive performance in a group of female industrial operators. Participants of the BF group improved their performance in attention, memory, and interference score, and significantly increased low frequencies—an index of baroreflex activity.

In our study, the intervention did not influence resting prefrontal blood oxygenation. Previous studies that administered a nirHEG intervention obtained similar results with healthy and ADHD children [59,60,82]. However, no studies involving healthy old adults used a combined intervention of HRV-BF and nirHEG-NF before; thus, we cannot compare our results or draw a general conclusion. In the present study, participants revealed significantly higher rest HEG ratio values at the median pole Fpz compared to the left frontal pole Fp1 and the right frontal pole Fp2. It is possible that the 2-min rest measurement (consisting in silently counting back from 500) was sufficient to induce mental fatigue, resulting in decreased activation of Fp1 and Fp2. However, during nirHEG training, the BF group exhibited larger HEG gains than the CTRL group, with a greater effect on the Fpz location. It is important to note that the use of nirHEG to assess the effectiveness of interventions aimed at improving blood flow oxygenation is still a subject of debate. This is because blood flow changes in the brain can be influenced by a variety of factors, including age, size, thickness of the skull, brain maturation, and the type of brain activity being performed. In the field of nirHEG interventions, previous research provided a relevant contribution to the understanding of such differences. For example, Serra-Sala et al. [83,84] showed that nirHEG signal could decrease when individuals process negative emotions, while it could increase during cognitive activity. In addition, the authors found that nirHEG could detect changes in how different age groups process information of different natures. In their study, adolescents but not young adults showed increased nirHEG activity during emotional-sensitivity processing. On the other hand, Pecyna and Porkorski [85] found that nirHEG could detect differences between subjects with dyslexia and healthy controls in a population of children and young adults. Still, it showed intervention-related changes only in the youngest group.

In the domain of NF interventions, there is evidence of the beneficial effect of brain wave NF (EEG-NF) training on the executive functioning of older adults. In the pioneer study of Becerra et al. [28], the authors found that healthy older adults were able to significantly increase attention and executive functions after NF training that targeted the reduction of theta waves (a frequency band that has proven to increase with normal aging and that is associated with greater cognitive impairment in patients with dementia [86]. Wang and Hsieh [87] also showed that NF training improved attention and working memory performance in healthy older adults. Among multidomain intervention research, the pioneer study of Meeuwsen et al. [37] proposed a program combining HRV-BF, NF, and other strategies to enhance psychological wellbeing and cognition in a population of older adults with objective and subjective memory complaints. The authors administered two different batteries targeting several domains (the MoCA and the NeuroTrax) to assess cognitive functions. They found significant improvements in the score of the MoCA, which evaluates seven cognitive functions (visuospatial/executive, attention, naming, language, delayed recall, abstraction, and orientation). They also reported significant improvements in self-reported measures of mental health and quality of life at the post-test. However, the program failed to elicit changes in HRV parameters, although participants could improve their breathing rate. Meeuwsen et al.’s study [37] has the merit of being the first to propose a multidomain program, combining HRV-BF and NF to assess changes in autonomous regulation, brain activation, cognitive functioning, and psychological wellbeing in older adults with memory impairments. Nevertheless, an important limitation was that the study did not include a control group but used a waitlist period preceding the pre-test that worked as the participant’s own control. Hence, it is difficult to prove that the observed improvements are a consequence of the intervention.

To the best of our knowledge, the current study is the first to systematically explore the combined effect of HRV-BF and nirHEG-NF on executive functions of older healthy adults. In addition, the present study can provide valuable insight into recruitment feasibility during a pandemic. In fact, it is important to note the challenges and disadvantages that may arise when implementing preventive measures to protect vulnerable populations in the context of laboratory-based research. There are a number of potential inconveniences that future researchers in this field should take into account: increasing number of dropouts; increasing anxiety and stigmatization in older adults; the limited possibility of replacing the sick experimenters, delivering the intervention at the scheduled time resulting in an increased delay between training sessions. A possible solution to avoid such pitfalls could rely on remotely smartphone-delivered biofeedback interventions. Recent findings have shown that these techniques can effectively decrease stress-related symptoms and depression, making them an attractive low-cost alternative, especially in times of pandemic emergency [88,89]. Thus, future research should explore the applicability of remote biofeedback on cognitive functioning.

## 5. Limitations and Outlook

This study was conducted overlapping with the COVID-19 pandemic, which made recruitment of older adults in large time periods virtually impossible. Consequently, the small sample size underpowered our results and failed to provide a meaningful between-group effect size estimate. Another drawback was using a control group too similar to the BF group. While the CTRL set-up was chosen to be comparable to the intervention group, just lacking the feedback loop, it is possible that participants in the control group may still have unconsciously modified their breathing rate following the pacer set at six breaths per minute, thus producing large-amplitude fluctuations HRV, resulting in less detectable differences between the two groups. Future studies should directly test this possibility by including an active control group dissimilar from the intervention group. For example, avoiding exposure of such control group to physiological instrumentation (potentially inducing placebo expectation) would limit an underestimation of the intervention benefits.

Further, the use of two tasks measuring the same construct (interference) but with different stimuli presentation (paper vs. computer) and response modality (oral vs. button-presses) may have weakened the consistency of our results. Meeuwsen et al.’s study [37] also showed that the administration of tasks in different modalities led to discrepant results between the severity of underlying symptoms measured by two different cognitive batteries. Thus, future research targeting several sub-components of a given domain or different cognitive domains should use the same modality of stimuli presentation to permit comparison of results. Finally, with regard to nirHEG results, this study found no evidence of a change in prefrontal blood oxygenation at rest. It is possible that the short duration of the training, added to the multiple sources of nirHEG variation mentioned in the above section, made it impossible to detect training-related changes in prefrontal blood oxygenation of older adults. However, further studies targeting an older population should administer longer training than 10 weeks. Also, we suggest that a follow-up should be necessary to estimate the maintenance of the intervention benefits at cognitive and physiological levels. In the present study, the incomplete record of the daily breathing exercises for participants of the BF group limited the reliability of their self-reported practice and made it impossible to control for this variable in our analyses. Given the crucial role that compliance plays in the effectiveness of an intervention and the achievement of desired outcomes, this limitation raises the question of how future research could design interventions that increase participant adherence to instructions and recommendations.

## 6. Conclusions

This study aimed to explore the impact of a combined intervention of HRV-BF and nirHEG-NF to increase inhibitory control in healthy older adults. Our findings indicate that this method possibly increases performance in attention/interference control under certain modalities, while the effect of the intervention on the inhibition of prepotent response needs to be further explored. Autonomous nervous system regulation and prefrontal blood oxygenation may also benefit from this intervention, as suggested by increased parasympathetic activity, overall HRV, and blood oxygenation in prefrontal areas during training. Previous literature showed that older adults make small changes in HRV parameters [90]. Our study proves that despite a very conservative control condition, the BF group increased HRV parameters above controls. Brain activation was also possible during training sessions. This study is the first to deliver a combined intervention that trains peripheral physiology and prefrontal blood oxygenation to enhance executive functions. HRV-BF and nirHEG-NF are two non-invasive, non-pharmaceutical, and low-cost methods that have the potential to be considered effective techniques to counteract cognitive decline and wellbeing in older populations. Due to the small sample size, this study was not powered to answer questions about efficacy. Instead, it can be considered a first attempt to evaluate the feasibility of the protocol and to provide valuable information about the potential mechanisms of efficacy for a new larger intervention. However, the rapid advances in neuroscience, machine learning, and computer engineering have led to the development of sophisticated techniques for the acquisition of brain and physiological signals. This makes it possible to develop comprehensive interventions merging BCI, BF, and NF approaches for rehabilitative and prevention purposes. For example, an intervention protocol could combine HRV training to increase self-regulatory capacity to improve health and general well-being; and BCI NF training to achieve an optimal cognitive level of functioning. However, more research is needed to investigate the specific benefits of this multimodal approach and to determine the best protocol for implementing it in a clinical and non-clinical setting.

## Figures and Tables

**Figure 1 brainsci-13-00335-f001:**
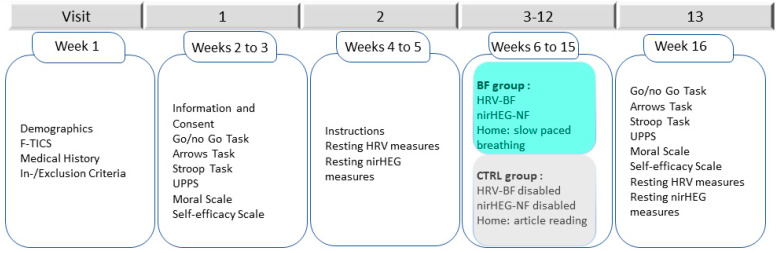
Experimental procedure: schedule of assessments for participants of both groups.

**Figure 2 brainsci-13-00335-f002:**
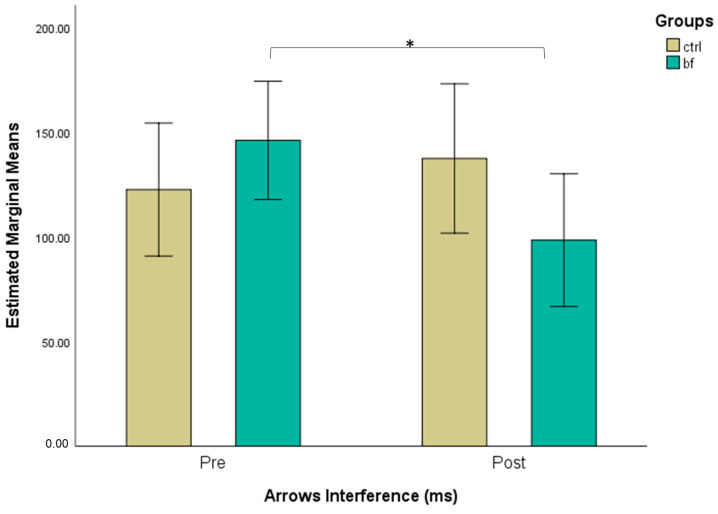
Estimated marginal means (with 95% confidence intervals) for Arrows Interference pre-and post-intervention across groups (*—*p* < 0.05—*t*-test for dependent samples).

**Figure 3 brainsci-13-00335-f003:**
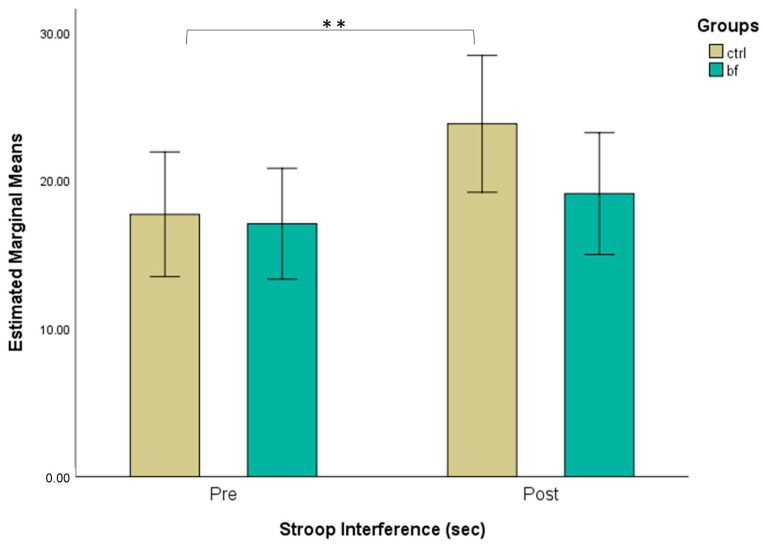
Estimated marginal means (with 95% confidence intervals) for Stroop Interference pre- and post-intervention across groups (**—*p* < 0.01—*t*-test for dependent samples).

**Figure 4 brainsci-13-00335-f004:**
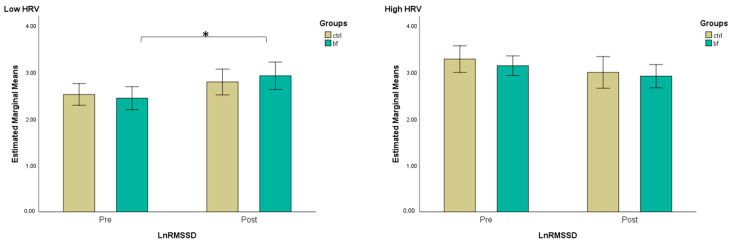
Estimated marginal means (with 95% confidence intervals) for ln RMSSD pre-and post-intervention across groups, separated by low and high RMSSD level at baseline (*—*p* < 0.05—*t*-test for dependent samples).

**Figure 5 brainsci-13-00335-f005:**
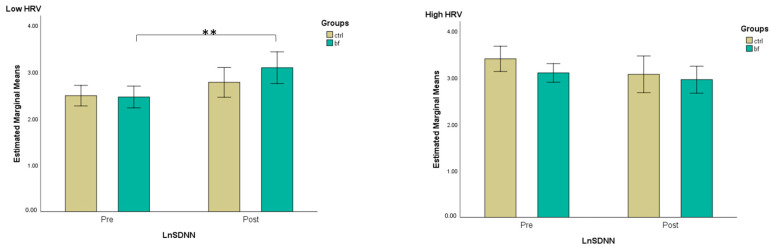
Estimated marginal means (with 95% confidence intervals) for ln SDNN pre-and post-intervention across groups, separated by low and high RMSSD level at baseline (**—*p* < 0.01—*t*-test for dependent samples).

**Figure 6 brainsci-13-00335-f006:**
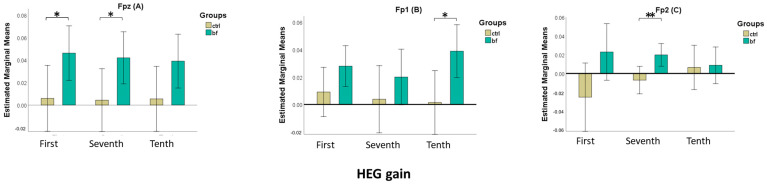
Bar chart showing the estimated marginal means (with 95% confidence intervals) for mean HEG gains by time session and by group, separated by site Fpz (**A**), Fp1 (**B**), and Fp2 (**C**) (*—*p* < 0.05, **—*p* < 0.01—F-test derived from one-way ANOVAs).

**Table 1 brainsci-13-00335-t001:** Descriptive statistics of demographic, cognitive, and physiological measures for each group at baseline.

	BF Group = 19 (15)		CTRL Group = 15 (11)	
Variable	Mean	SD	Mean	SD
Age (years)	70.78	3.97	70.92	4.33
% Female	78.9%		73.3%	
Education (years)	14.89	3.59	15.93	4.27
F-Tics	35.63	3.38	36.53	3.66
BMI (kg/m^2^)	23.81	4.29	23.44	4.53
PA	4.86	3.30	5.10	4.15
Arrows interference RTs (ms)	145.96	64.61	122.46	54.39
Go RTs (ms)	446.67	53.48	441.04	50.64
No Go commissions	1.95	1.54	2.40	1.80
Stroop interference RTs (sec)	17.07	7.70	17.70	8.36
LnRMSSD	2.84	0.47	2.82	0.52
LnSDNN l	2.83	0.49	2.85	0.51
HR (bpm)	74.35	11.83	72.37	9.97
Resp (Hz)	0.26	0.06	0.24	0.05
Fpz ratio	108.06	26.68	115.04	23.14
Fp1 ratio	96.74	22.99	93.86	12.56
Fp2 ratio	98.22	22.92	90.20	11.29

Note. Education = education in years; F-TICS = raw score; BMI = Body Mass Index raw score; PA = physical activity in hours per week; Arrows RTs and Go RTs are in msec; No Go commissions = numbers of errors; Stroop RTs are in sec; RMSSD = the root mean square of differences of successive RR intervals are in msec and natural log transformed; SDNN = standard deviation of all NN intervals are in msec and natural log transformed; HR = heart rate in beats per minute; Resp Hz = respiration rate are in cycles per sec; Fpz, Fp1, and Fp2 ratio = ratio between the red light and the infrared light × 100.

**Table 2 brainsci-13-00335-t002:** Pre- and post-means and standard deviations (SD) of nirHEG ratio by site and by group.

	BF Group (*N* = 18)		CTRL Group (*N* = 14)	
Variable	Mean	SD	Mean	SD
HEG ratio				
Fpz				
Pre	108.06	26.68	115.04	23.14
Post	113.93	30.38	117.98	26.73
Fp1				
Pre	96.74	22.99	93.86	12.56
Post	98.98	27.44	96.55	15.20
Fp2				
Pre	98.22	22.91	90.20	11.29
Post	100.29	28.79	94.29	13.91

**Table 3 brainsci-13-00335-t003:** Between-group differences in nirHEG percentage values during training sessions for each site and by group.

	BF Group			CTRL Group					
Variable	Mean	SD	*N*	Mean	SD	N	F	*p*	$\eta_{p}^{2}$
1st session									
Fpz	4.14	4.59	19	0.86	4.12	14	4.49	0.04	0.13
Fp1	2.67	2.57	18	0.91	2.94	14	3.25	0.08	0.09
Fp2	2.39	5.33	19	−1.51	5.73	14	4.05	0.05	0.11
7st session									
Fpz	3.71	5.18	18	0.24	3.10	13	4.62	0.04	0.14
Fp1	2.73	3.72	19	0.79	4.40	14	1.88	0.18	0.06
Fp2	1.88	2.35	19	−0.71	2.61	14	8.97	0.00	0.22
10th session									
Fpz	3.69	5.28	19	1.44	5.39	15	1.49	0.23	0.04
Fp1	3.75	3.98	18	0.57	2.97	15	6.51	0.02	0.17
Fp2	1.39	4.52	19	1.15	3.01	15	0.03	0.86	0.00

## Data Availability

The data presented in this study will be openly available in the Open Science Framework as soon as possible.

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
