# Peer review of "Enhancing Inhibitory Control in Older Adults: A Biofeedback Study"

_brainsci, 2023, doi:10.3390/brainsci13020335_

Round 1

Reviewer 1 Report

Interesting study. In general, it was well written and executed. The authors appropriately spell out the limitations, especially the difficulty in obtaining older participants. The findings are promising. A few suggestions:

1) Report the HRV measures in the original form (untransformed). I would defer to one of the co-authors, but we usually see the natural log used for transformations. Also, why wasn't ECG used for the baseline and follow-up resting measurements. That would have allowed Frequency Domain measures.  

2) The HEG method is much more controversial than discussed here. While blood flow is clearly being measured, whether changes from the surface reflect cortical blood flow has really not been researched. This may be why those measures did not seem to correlate with the important inhibition measures. 

3) why not report the correlations between RMSSD changes and the improvements in the interference tasks, especially for the BF group. 

4) There was little mention of home practice. Daily practice may be more important than in session procedures. Any information on this?

Overall, very relevant, interesting study. 

Reviewer 2 Report

The study aimed to explore the combined effect of HRV-BF and nirHEG-NF on executive functions of older healthy adults.

An undeniable merit of the work is the application of a multimodal biofeedback intervention to improve inhibitory control.

The Introduction reflects the state of the chosen research topic, but I think it is worth adding an analysis of articles on the topic of the effect of different types of biofeedback on brain functioning. For example, on functional activity and asymmetry of hemispheres. Executive functions are realized by the brain through distributed brain activity.

Also, more could also be added to the Introduction about the brain aspect of the realization of executive functions and inhibitory control in particular. When the authors add these aspects to the Introduction section, they can also diversify the Discussion section.

The materials and methods of the study and the Results are described clearly and correctly.

P.S. The authors should check for typos and unnecessary word hyphenation. For example, in the abstract we see "Thir-ty-four participants".

Reviewer 3 Report

Undoubtedly, multidomain interventions based on bio-/neurofeedback have proven useful in improving executive functions. Authors in this paper aimed to explore the potential efficacy and feasibility of an intervention that combined Heart Rate Variability Biofeedback and Near Infrared Hemoenceph-alography Neurofeedback on inhibitory control of healthy older adults. 

What is important, besides cognitive outcomes, the study examined pre-post changes in autonomic regulation and prefrontal blood oxygenation at rest and during training. Results revealed training-induced inhibitory control gains in one of the two interference tasks, whereas no effect was found on response inhibition. 

My comments on the article are as follows:

- I suggest you consider adding to your keywords: Inhibitory control

- I propose to extend the introduction to the article by referring to the BCI technology, the development of which is important for the implementation of neurofeedback. For example, you can refer to the publication: Brain–Computer Interface Technology, Studies in Computational Intelligence, Springer, Volume 852, Pages 11 - 17, 2020, doi: 10.1007/978-3-030-30581-9_3.

- I am asking you to support the choice of these and not other statistical methods for data analysis.

- As part of the Conclusions, I propose to describe plans for the future in the field of this research.
